# Comorbidities at Diagnosis, Survival, and Cause of Death in Patients with Chronic Lymphocytic Leukemia: A Population-Based Study

**DOI:** 10.3390/ijerph18020701

**Published:** 2021-01-15

**Authors:** Alicia Villavicencio, Marta Solans, Lluís Zacarías-Pons, Anna Vidal, Montse Puigdemont, Josep Maria Roncero, Marc Saez, Rafael Marcos-Gragera

**Affiliations:** 1Epidemiology Unit and Girona Cancer Registry, Oncology Coordination Plan, Catalan Institute of Oncology, Carrer del Sol 15, 17004 Girona, Spain; marianna.vidal@iconcologia.net (A.V.); mpuigdemont@iconcologia.net (M.P.); rmarcos@iconcologia.net (R.M.-G.); 2Research Group on Statistics, Econometrics and Health (GRECS), University of Girona, C/Universitat de Girona 10, 17003 Girona, Spain; marta.solans@udg.edu (M.S.); marc.saez@udg.edu (M.S.); 3Descriptive Epidemiology, Genetics and Cancer Prevention Group, Biomedical Research Institute (IDIBGI), 17190 Girona, Spain; 4Centro de Investigación Biomédica en Red: Epidemiología y Salud Pública (CIBERESP), Av. Monforte de Lemos 3-5, 28029 Madrid, Spain; 5Research Group on Aging, Disability and Health, Girona Biomedical Research Institute (IDIBGI), 17190 Girona, Spain; lzacarias@idibgi.org; 6Hematological Service, Josep Trueta University Hospital, Catalan Institute of Oncology, Avinguda de França, S/N, 17007 Girona, Spain; jroncero@iconcologia.net; 7Josep Carreras Leukemia Research Institute, 08916 Badalona, Spain

**Keywords:** chronic lymphocytic leukemia, comorbidities, causes of death, survival, population-based

## Abstract

This study aimed to examine the prevalence of comorbidities in patients diagnosed with chronic lymphocytic leukemia (CLL), and to assess its influence on survival and cause-specific mortality at a population-based level. Incident CLL cases diagnosed in the Girona province (Spain) during 2008–2016 were extracted from the Girona Cancer Registry. Rai stage and presence of comorbidities at diagnosis, further categorized using the Charlson comorbidity index (CCI), were obtained from clinical records. Observed (OS) and relative survival (RS) were estimated and Cox’s proportional hazard models were used to explore the impact of comorbidity on mortality. Among the 400 cases included in the study, 380 (99.5%) presented at least one comorbidity at CLL diagnosis, with diabetes without end organ damage (21%) being the most common disease. 5-year OS and RS were 68.8 (95% CI: 64.4–73.6) and 99.5 (95% CI 3.13–106.0), respectively, which decreased markedly with increasing CCI, particularly in patients with CCI ≥ 3. Multivariate analysis identified no statistically significant association between the CCI and overall CLL-related or CLL-unrelated mortality. In conclusion, a high CCI score negatively influenced the OS and RS of CLL patients, yet its effect on mortality was statistically non-significant when also considering age and the Rai stage.

## 1. Introduction

Chronic lymphocytic leukemia (CLL) is the most common leukemia in Western countries, with an incidence of 3.79 [1] and a 5-year relative survival (RS) of 69 years (95% confidence interval (CI):68.1; 69.8) [2], in 2006–2008 in Europe. Although CLL is often classified as an indolent disease with a relatively good prognosis, it has an unpredictable clinical course and can become resistant to conventional treatments. Despite recent progress in its management, this disease remains incurable, and patients with CLL still have a reduced life expectancy as compared to the general population [3].

CLL typically occurs in advanced ages (its median age at diagnosis is >70 years) [1,4]. Elderly patients are often compromised by concurrent pathological conditions [5], and particularly in cancer, comorbidities are a significant concern (i.e., they preclude some treatments or are a competing cause of death) [6]. Indeed, cumulative evidence reflects that the overall survival of cancer populations decreases as the burden of comorbid diseases increases [7]. The impact of comorbidities on CLL outcomes, however, remains less explored. Several studies assessed the impact of comorbidities on CLL survival/mortality [8,9,10,11,12,13,14,15] and treatment tolerance or feasibility [10,16,17,18]. However, few include population-based data [12,15,19], and thus, there is a lack of real-word data addressing CLL outcomes in comorbid patients, especially when considering the specific causes of death.

Therefore, the aim of this study was to examine the prevalence of comorbidities and their influence on survival and mortality (overall and CLL-related or CLL-unrelated) of patients diagnosed with CLL, in the province of Girona (Spain), during 2008–2016.

## 2. Materials and Methods

### 2.1. Study Population

Incident CLL cases diagnosed during the period of 2008–2016 were extracted from the population-based Girona Cancer Registry (GCR). The GCR is located in the Northeast of Catalonia, in Spain, covering a population of 739,607 inhabitants in 2016. Following WHO recommendations, CLL and small lymphocytic lymphoma (SLL) cases were classified together, since both share clinical and pathological features [20]. We excluded death certificate only (DCOs) and those cases diagnosed at autopsy.

### 2.2. Comorbidity Assessment

Data on the Rai stage, indicating severity of CLL and comorbidities present at diagnosis were retrospectively obtained by reviewing the medical records. Comorbidities were assigned to one of the following categories—acute myocardial infarction, congestive heart failure, peripheral vascular disease, cerebrovascular disease, dementia, chronic lung disease, rheumatic disease, peptic ulcer, mild liver disease, mild/moderate diabetes, diabetes with chronic complications, hemiplegia/paraplegia, kidney disease, malignant tumors, moderate/serious liver disease, metastatic tumor, and AIDs [14]. The Charlson comorbidity index (CCI) [21,22] was calculated for each patient, based on the health conditions present at the time of diagnosis. The CCI is a prognostic tool based on the principle that age and the presence and severity of comorbidities increase the likelihood of mortality among patients who receive treatment for chronic illnesses. Based on their CCI scores, patients were grouped into five groups (i.e., absence of comorbidity (0), low risk (1–2), moderate risk (3), high risk (>4), and unknown CCI status).

### 2.3. Survival and Mortality Data

Patients were followed up until death or last follow-up date (31 December 2019), whichever came first. Data on the vital status of patients were obtained by linking records to the Catalan Registry of Mortality and the National Death Index [23]. Mid-year population estimates and mortality rates in the Girona province were obtained from the Institut d’Estadística de Catalunya, IDESCAT [24].

In our region, the cause of death was initially determined by the treating physician, based on the available clinical information, and sometimes on autopsy reports. This information was then transferred to the Catalan Registry of Mortality Data [25], which was responsible for coding the underlying causes of death (basic cause of death), following the guidelines of the International Classification of Diseases, 10th edition (CIE-10-ES) [26]. In our study, causes of death were categorized into CLL-related (including all hematological malignancies) and CLL-unrelated death (Appendix A). Those patients with unknown cause of death (*n* = 13) were excluded for both the CLL-related and CLL-unrelated survival analyses.

### 2.4. Statistical Analysis

Descriptive statistics were used to summarize the baseline clinical characteristics, overall and by the CCI score and cause of death. Differences in the clinical characteristics by CCI score and cause of death were assessed by the chi-square test. Observed survival (OS) was modelled using the Kaplan-Meier method. The relative survival (RS) rates were estimated using the Pohar–Perme method [27] as the ratio of observed survival in the study population, to expected survival in the general population of the same age, sex, year, and province (Girona) [28]. Expected survival rates were taken from the life tables for the population covered by the GCR. Comparison of OS and RS curves were performed using a log-rank type test [29]. To assess the effect of CCI score after adjusting for other covariates (gender, age, Rai stage, and period of diagnosis), multifactor Cox proportional hazards models were constructed, and a Wald test was used. The adjusted hazard ratios of death (HR) and the corresponding 95% confidence intervals (95% CI) were estimated. For all analyses, a *p*-value <0.05 was considered to be significant. All analyses were performed using the free software R (RStudio version 1.1.463) (Free Software Foundation, Inc., Boston, MA, USA).

## 3. Results

### 3.1. Prevalence of Comorbidity

A total of 400 incident CLL with a median follow-up time of 5.2 years were included in the study. Clinical characteristics of patients at diagnosis, overall and by CCI score, are detailed in Table 1. Median age at diagnosis was 72 years (interquartile range, 60–80 years); there were 230 (57.5%) males, and most patients (56.7%) had early stage CLL (Rai stage 0) at diagnosis. As expected, patients with higher CCI score were older and more prone to be diagnosed with advanced Rai stages, while we did not find differences according to sex or period of diagnosis. Except for 20 patients (0.5%), the population studied presented at least one comorbidity at the time of diagnosis (Table 1). Among the comorbid patients, the CCI score was low (1–2) in 86 (21.5%), moderate (3–4) in 151 (37.7%), high (>4) in 118 (29.5%), and unknown in 25 (6.2%).

At diagnosis, 21% patients had diabetes, 18% had congestive heart failure, and 11% of patients had malignant tumors and chronic lung disease. Figure 1 shows the prevalence of comorbidities by sex. In general, diabetes without end-organ damage, congestive heart failure, cancer, chronic lung disease, and dementia were the most frequent comorbidities, being more predominant in men (except for dementia, which was more frequent in women). Additionally, in our cohort, 5 (1.25%) patients with CLL progressed to aggressive lymphoma (Richter syndrome).

### 3.2. Comorbidity and Survival

1, 3, and 5-year OS and RS survival of CLL patients, according to the CCI is displayed in Table 2. Overall, 5-year OS and RS were 68.8 (95% CI: 64.6–73.6) and 99.5 (95% CI: 93.6–106.0), respectively. Survival estimates decreased markedly with increasing CCI scores, particularly in patients with 3 or more comorbidities.

Figure 2 further depicts these differences among the OS curves (*p*-value of log rank test <0.001; also for RS survival curves—data not shown).

### 3.3. Comorbidity and Mortality

Among the 168 patients that died during the follow-up, the cause of death could be accurately determined in 155 (92.2%), 86 (55.5%) being CLL-related (Appendix A). The distribution of clinical features according to the causes of death is shown in Appendix A. No statistically significant differences were observed according to CCI, age, sex, Rai stage, or period of diagnosis. Table 3 examines the relationship between the CCI score and other clinical variables with mortality. On univariate analysis, a higher CCI score was associated with higher overall mortality (HR: 5.82; 95% CI: 1.44–23.49), but not with CLL-related (HR: 1.28; 95% CI: 0.17–9.18) or and CLL-unrelated (HR: 2.83; 95% CI: 0.39–20.72) mortality. On multivariate analysis, after adjusting for age, sex, Rai stage, and period of diagnosis, a higher CCI score was not associated with a higher overall mortality (HR: 2.05; 95% CI: 0.47–8.90) and CLL-related mortality (HR: 0.62; 95% CI: 0.07–5.30).

## 4. Discussion

This population-based study provides prevalence data of comorbidities in CLL patients and evaluates its impact, through the CCI score, on survival and mortality in the Girona province, over a period of 8 years. Our results indicate that almost all patients had, at least, one comorbidity at CLL diagnosis, and that the 5-year OS and RS decreased markedly with higher CCI scores, particularly with CCI ≥ 3. However, in multivariate analysis, a higher CCI score was not related to an increased mortality (both overall, CLL-related, and CLL- unrelated).

The association between comorbidities and poorer outcomes in cancer patients is well established [7], yet specific data on CLL are scarce. To the best of our knowledge, two clinical trials [10,30], one prospective cohort [14], two population-based [12,15], two multicenter [11,31], and three hospital series [8,9,13] assessed the relationship between comorbidities at diagnosis and survival or mortality in CLL patients. In line with our results, most studies evidenced lower survival rates in patients with a higher level of comorbidities [8,9,10,12,13,14], yet there were mixed results regarding their independent effect on survival or mortality [10,12,13,14,30]. However, comparisons must be made with caution, especially because there were different study designs and settings. This could be evidenced when comparing the median age of study populations, which was lower (≤65 years) in the clinical trials [10,30] or prospective cohorts [14]. This might have influenced the outcomes of patients, not only if different treatment patterns were used, but also if the prevalence and type of comorbidities at diagnostic varied within them. Nevertheless, most authors found, in line with our study, that the most frequent comorbidities at CLL diagnosis were diabetes mellitus, congestive heart failure, and chronic lung disease [8,10,15,30,32]. It is likely that a higher prevalence of diabetes mellitus is related to these patients having continuous check-ups and therefore the finding of CLL is incidental. Indeed, among the 65 patients with diabetes mellitus, 45 had a low Rai (0–1), which suggests that if these patients were under continuous control and follow-up, CLL could probably be diagnosed at an early stage.

To date, no comorbidity score was prospectively validated in CLL, and thus, there is a marked heterogeneity in the study methodology when assessing comorbidities in such patients. For instance, different range of disease codes were considered, different scales or scoring systems were used (i.e., number of comorbidities [8,15,30] vs. CCI [10,14] vs. Cumulative Illness Rating Scale [9,11,12,13,31]), and within the CCI, different diseases codes were included and final score groupings were used. Overall, this reinforced the need to standardize the assessment of comorbidity in patients with CLL.

Our research is one of the few to explore the role of comorbidities on mortality in patients with CLL, considering the specific cause of death. Previous studies yielded mixed results, indicating that comorbidities might affect the overall mortality [10,12,13,14], CLL-related [10,15] and CLL-unrelated [14,30]. In our multivariate analysis, after adjusting for age and Rai stage, we did not find a statistically significant association between the CCI and mortality (overall or both when considering CLL-related or CLL-unrelated causes). However, our results must be interpreted with caution, since we might have been limited by sample size and, particularly, by death cause misclassification [33]. We relied on official data on the basic cause of death—and not the secondary cause of death [25]. Thus, we were unable to subclassify CLL-related mortality into more informative categories (e.g., disease progression, second primary cancer, infection, and other health conditions). In CLL-related causes, we included all hematological malignancies, suspecting misclassifications of CLL cases and treatment-related diseases. However, there might be some real secondary primary hematological malignancies that were wrongly located in CLL-related causes.

To date, several prognostic tools were developed to predict outcomes of CLL patients [34]. Among them, the CLL-IPI is prognostic index most widely used [35] that has been validated in an elderly population [36]. Although overall survival is currently undergoing significant changes with the introduction of novel agents [3], most still hold the potential to support clinical patient management. However, due to drug interactions and a different side effect profile, specific comorbidities and comedication might have a larger impact on survival in the future [34]. Overall, further real-world data including clinical and treatment variables are warranted to deepen into the role of comorbidities on CLL outcomes, and to provide better-tailored prognostic tools for the comorbid and elderly populations.

Several limitations must be considered when interpreting our results, including the limited sample size and, particularly, potential misclassification or categorization of the underlying cause of death [33], as previously discussed. Similarly, clinical data were gathered from electronic health records, which cannot be assumed to provide complete, accurate, and standardized information of individual’s health status. In addition, our study evaluated comorbid health conditions only at diagnosis; however, during follow-up, some patients might acquire new comorbid conditions or face a decline in their organ function. Furthermore, we lacked data on additional variables that could influence the prognosis, such as treatment patterns or the presence of several genetic or biochemical markers (e.g., TP53 dysfunction or a complex karyotype) [34]. Finally, race/ethnicity was not recorded in our study, which might also influence survival rates, as previously reported [37].

## 5. Conclusions

Pre-diagnostic comorbidities are extremely common in CLL patients. Survival estimates decrease markedly with higher CCI scores, especially in patients with a CCI score ≥3. However, comorbidities are less important than age and stage in predicting mortality (both CLL-related or CLL-unrelated) in newly diagnosed patients with CLL. These population-based data will provide insights into the relationship between comorbidities and CLL in a real-world setting. Prioritizing comorbid CLL patients in future clinical trials is warranted, to further inform treatment guidelines and improve outcomes for these patients.

## Figures and Tables

**Figure 1 ijerph-18-00701-f001:**
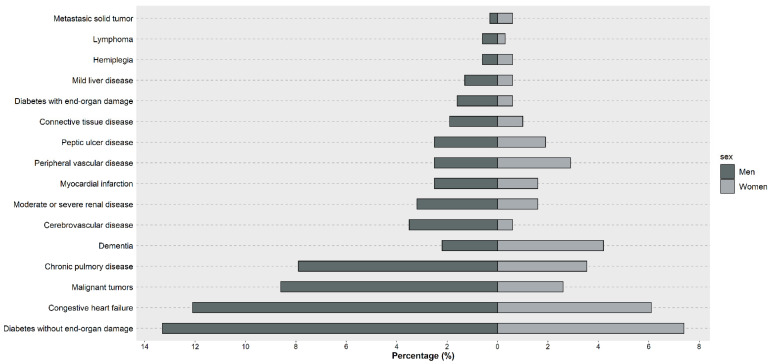
Frequency by sex of comorbidities included in the Charlson comorbidity index among patients with chronic lymphocytic leukemia in Girona, Spain (*n* = 400).

**Figure 2 ijerph-18-00701-f002:**
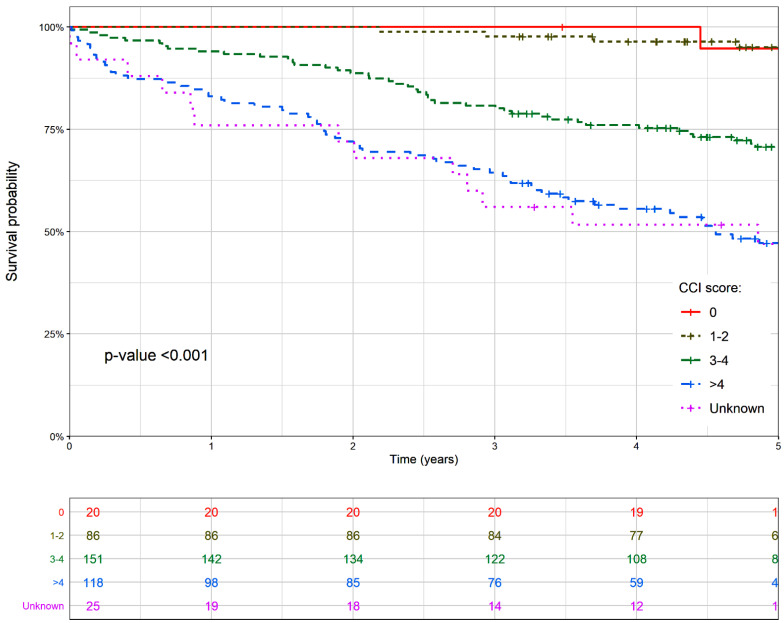
Observed survival of patients with chronic lymphocytic leukemia by the Charlson comorbidity index at diagnosis. Events included both types of death, CLL-related and CLL-unrelated. The differences between the survival curves that had a *p*-value <0.05 were considered to be statistically significant.

**Table 1 ijerph-18-00701-t001:** Distribution of baseline characteristics of chronic lymphocytic leukemia patients according to the Charlson comorbidity index score.

Clinical Characteristics		Charlson Comorbidity Index (CCI)					
	Total	Absence of Comorbidity	Low Risk	Moderate Risk	High Risk	Unknown	
		0	1–2	3–4	>4		*p*-Value
	*n* (%) a	*n* (%) a	*n* (%) a	*n* (%) a	*n* (%) a	*n* (%) a	
**Total**	400 (100.0)	20 (5.0)	86 (21.5)	151 (37.7)	118 (29.5)	25 (6.2)	
**Age**							
Mean (SD)	70.8 (12.2)	43.6 (5.5)	59.41(5.6)	73.11 (7.2)	80.58 (7.7)	70.96 (11.8)	<0.001
Median (Range)	72 (62–80)	46 (39.7–48)	59 (55–63.7)	74 (69–78)	83 (75.2–86)	69 (60–79)	
**Gender**							
Male	230 (57.5)	9 (45.0)	46 (53.5)	84 (55.6)	74 (62.7)	17 (68.0)	0.342
Female	170 (42.5)	11 (55.0)	40 (46.5)	67 (44.4)	44 (37.3)	8 (32.0)	
**Age group**							
<65	122 (30.5)	20 (100.0)	65 (75.6)	22 (14.6)	5 (4.2)	10 (40.0)	<0.001
65–78	158 (39.5)	-	21 (24.4)	97 (64.2)	32 (27.1)	8 (32.0)	
>78	120 (30.0)	-	-	32 (21.2)	81 (68.6)	7 (28.0)	
**Rai stage**							
0	227 (56.7)	12 (60.0)	57 (66.3)	85 (56.3)	68 (57.6)	5 (20.0)	<0.001
I-II	60 (15.0)	5 (25.0)	15 (17.4)	29 (19.2)	11 (9.3)	-	
III-IV	35 (8.7)	2 (10.0)	6 (7.0)	12 (7.9)	15 (12.7)	-	
Unknown	78 (19.5)	1 (5.0)	8 (9.3)	25 (16.6)	24 (20.3)	20 (80.0)	
**Period of diagnostic**							
2008–2010	141 (35.2)	6 (30.0)	27 (31.4)	54 (35.8)	45 (38.1)	9 (36.0)	0.320
2011–2013	136 (34.0)	12 (60.0)	27 (31.4)	51 (33.8)	37 (31.4)	9 (36.0)	
2014–2016	123 (30.7)	2 (10.0)	32 (37.2)	46 (30.5)	36 (30.5)	7 (28.0)	

a Except when specified.

**Table 2 ijerph-18-00701-t002:** Observed and relative survival of chronic lymphocytic leukemia patients according to the Charlson comorbidity index at diagnosis in Girona, 2008–2016.

CCI Core	1-Year		3-Years		5-Years	
	OS (95% CI)	RS (95% CI)	OS (95% CI)	RS (95% CI)	OS (95% CI)	RS (95% CI)
0	100.0 (100.0–100.0)	100.0 (100.0–100.0)	100.0 (100.0–100.0)	100.0 (100.0–100.0)	94.7 (85.2; 100.0)	95.4 (86.1; 106.0)
1–2	100.0 (100.0–100.0)	100.0 (100.0–100.0)	97.7 (94.5; 100.0)	99.7 (96.5; 103.0)	95.1 (90.4; 99.9)	98.8 (94.0; 104.0)
3–4	94.0 (90.3; 97.9)	96.5 (92.7; 100.5)	80.8 (74.7; 87.3)	88.1 (81.2; 95.6)	70.7 (63.7; 78.5)	82.9 (73.5; 93.4)
>4	83.0 (76.5; 90.1)	89.3 (82.3; 96.9)	64.4 56.3; 73.6)	82.2 (71.6; 94.3)	47.2 (38.7; 57.5)	72.2 (57.7; 90.5)
Unknown	76.6 (61.0; 94.7)	81.2 (66.1; 99.9)	56.0 (39.6; 79.3)	62.6 (44.1; 88.7)	47.0 (30.7; 71.9)	52.7 (33.8; 82.3)
Overall	91.2 (88.5; 94.1)	104.0 (104.0–104.0)	79.0 (75.1; 83.1)	113.0 (113.0–113.0)	68.8 (64.4; 73.6)	99.5 (93.6; 106.0)

CCI—Charlson comorbidity index; OS—observed survival; RS—relative survival; 95% CI, 95% confidence interval.

**Table 3 ijerph-18-00701-t003:** Univariate and multivariate analysis of mortality (overall, CLL-related, and CLL-unrelated) in patients with chronic lymphocytic leukemia in Girona (2008–2016).

Variable			Univariate Analysis						Multivariate Analysis					
	*N*		Overall Mortality (*n* = 400)		CLL-Related Mortality (*n* = 86)		CLL-Unrelated Mortality (*n* = 69)		Overall Mortality (*n* = 400)		CLL-Related Mortality (*n* = 86)		Unrelated to CLL Mortality (*n* = 69)	
			HR (95% CI)	*p*-Value	HR (95% CI)	*p*-Value	HR (95% CI)	*p*-Value	HR (95% CI)	*p*-Value	HR (95% CI)	*p*-Value	HR (95% CI)	*p*-Value
**Sex**	Female	170	1.00		1.00		1.00		1.00		1.00		1.00	
	Male	230	1.28 (0.94; 1.75)	0.118	0.85 (0.54–1.31)	0.457	1.11 (0.68–1.81)	0.680	1.38 (1.01; 1.90)	0.042	0.81 (0.47; 1.39)	0.447	1.18 (0.70–2.01)	0.529
**Age (y)**	<65	122	1.00		1.00		1.00		1.00		1.00		1.00	
	65–78	158	8.45 (5.18; 13.78)	<0.001	2.53 (1.17–5.46)	0.018	1.42 (0.60–3.34)	0.426	2.72 (1.59; 4.66)	<0.001	2.16 (0.92;5.09)	0.078	1.54 (0.57–4.16)	0.391
	>78	120	2.75 (1.65; 4.58)	<0.001	2.78 (1.34–5.78)	0.006	1.37 (0.60–3.15)	0.455	9.96 (5.86; 16.92)	<0.001	2.20 (0.94; 5.13)	0.068	2.05 (0.76–5.51)	0.157
	p-trend ^1^			<0.001		<0.001		0.07		<0.001		0.015		0.396
**Rai stage**	0	227	1.00		1.00		1.00		1.00		1.00		1.00	
	I-II	60	1.49 (0.97; 2.31)	0.067	0.61 (0.34–1.09)	0.094	0.66 (0.28–1.59)	0.361	1.91 (1.23–2.96)	0.004	0.75 (0.39; 1.41)	0.366	1.13 (0.44–2.91)	0.804
	III-IV	35	2.68 (1.68; 4.27)	<0.001	1.04 (0.55–1.97)	0.901	1.83 (0.87-3.84)	0.111	4.17 (2.56; 6.80)	<0.001	1.18 (0.57; 2.44)	0.650	8.24 (3.26–20.83)	<0.001
	Unknown	78	2.17 (1.47; 3.18)	<0.001	1.70 (0.96–3.02)	0.069	1.71 (0.95–3.06)	0.072	1.95 (1.25; 3.04)	0.003	1.82 (0.79; 4.22)	0.159	1.56 (0.79–3.09)	0.202
	p-trend ^1^			0.002		0.06		0.2		<0.001		0.428		<0.001
**Period of diagnosis**	2008–2010	141	1.00		1.00		1.00		1.00		1.00		1.00	
	2011–2013	136	1.02 (0.73; 1.42)	0.900	1.15 (0.72–1.84)	0.567	2.77 (1.52–5.04)	<0.001	1.35 (0.96; 1.91)	0.088	1.13 (0.66–1.95)	0.651	3.71 (1.95–7.07)	<0.001
	2014–2016	123	0.50 (0.31; 0.81)	0.004	1.80 (0.86–3.76)	0.121	5.06 (2.33–10.98)	<0.001	0.65 (0.39; 1.07)	0.092	1.04 (0.37–2.88)	0.946	9.95 (4.10–24.12)	<0.001
	p-trend ^1^			0.1		0.08		0.003		0.765		0.003		<0.001
**CCI**	0	20	1.00		1.00		1.00		1.00		1.00		1.00	
	≥1	355	5.82 (1.44-23.49)	0.013	1.28 (0.17–9.18)	0.815	2.83 (0.39–20.72)	0.305	2.05 (0.47–8.90)	0.340	0.62 (0.07–5.30)	0.660	8.67 (0.87–85.96)	0.065
	Unknown	25	10.08 (2.30-44.42)	0.002	1.81 (0.22–14.55)	0.576	8.34 (0.89–78.36)	0.063	4.11 (0.85–19.90)	0.078	0.50 (0.05–5.20)	0.560	68.08 (4.73–980.51)	0.002
**CCI continuous**		375	1.45 (1.34-1.57)	<0.001	1.19 (1.04–1.37)	0.01	1.21 (1.05–1.39)	0.01	1.16 (1.01–1.33)	0.034	0.96 (0.77–1.20)	0.717	1.27 (1.06–1.53)	0.01

CLL—chronic lymphocytic leukemia; 95% CI, 95% confidence interval; CCI—Charlson comorbidity index; HR—hazard ratio. ^1^
*p*-value of the Cox proportional model fitted with the ordinal variable as continuous to test for the lineal trend (the unknown category was excluded from the calculation).

## Data Availability

Data are contained within the article or Appendix A.

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
