# Peer review of "Comorbidities at Diagnosis, Survival, and Cause of Death in Patients with Chronic Lymphocytic Leukemia: A Population-Based Study"

_ijerph, 2021, doi:10.3390/ijerph18020701_

Round 1

Reviewer 1 Report

Authors described the association between comorbidities and survival outcomes in patients  with chronic lymphocytic leukemia (CLL).

  1. Comorbidities was not correlated to mortality, while Rai stage was correlated. Nevertheless, CLL-unrelated mortality was related. It is why?
  2. What causes were these patients death?

Reviewer 2 Report

This study evaluates the prognostic role of Charlson comorbility index (CCI) in chronic leukemia lymphocytic leukemia. The following comment is raised:

  1. What is the ethnic composition of your cohort? Please discuss the potential role of ethnicity in influencing outcomes.

Reviewer 3 Report

The authors present a retrospective cohort from the Girona cancer registration and report on the prevalence of comorbidities and its influence on overall survival, relative survival, CLL-related and -unrelated survival in a period of 2008-2016. The overall work is performed well, however a number of issues needs to be addressed.

Major comments

In this study, all patients that died due to hematological malignancies are classified as CLL-related death. CLL patients have an increased risk of the developing secondary primary cancers (hematological and non-hematological). Although the nature of association with second cancer in CLL is not completely clear yet, several possible mechanisms have been suggested such as the immunodeficiency associated with disease and chemo-immunotherapy. However, the authors did not report on previous therapies in these patients. Furthermore, they did not report whether the treating physician regarded this hematological malignancy as a secondary malignancy that was caused by the CLL(treatment). As such, these other hematological malignancies can also a result of coincidence. The authors should include information about receive therapy and/or should discuss this thoroughly in the discussion section. Furthermore, CLL can transform to an aggressive lymphoma which could increase the likelihood of death. However, the authors did not report on the incidence of Richter transformation in these patients. Moreover, patients with CLL tend to have disfunction of the T-cell which results in an impaired immunity. As such, CLL patients suffer from an increased incidence of infections which could also contribute to death. However, the category of infections in the CLL-related cause of death is lacking (Supplemental Table S1). I suggest the author either include this to CLL-related death or change the classification to one that is more informative (i.e., CLL progression, second primary cancer, infection, other health conditions).

Relative survival is estimated by taking account of the characteristics of the general population (the same period, sex, and age group). Method section lacks the information regarding RS, can the authors please include how RS was calculated and whether they performed matching for age, sex, and time period? Can the authors comment how these two measurements are related? Since relative survival reflects the survival of the general population, can it be stated that the proportion of excess mortality is due to CLL-related death?

The authors reported that the most frequent comorbidity at diagnosis was diabetes without end organ damage (21%). The diagnosis of CLL is nowadays commonly based on a lymphocytosis that was found during routine blood examination. As such, it could be suggested that it would be logical that diabetes was present in many patients since diabetic patients need to undergo regular blood check-ups. This could be reflected by a higher number of low Rai (0-1) patients. Can the authors report whether they also saw this phenomenon in their patient population? If so, please discuss this in the discussion section.

Minor points

Table 2: please include all the confidence intervals.

Table 3: please include p-trend for variables having >2 categories.

Line 89-90: “Those patients with unknown cause of death were excluded for both the CLL-related and CLL-unrelated survival analyses”: please provide numbers of excluded patients

Line 109-110 “Except for…at diagnosis”: The sentence isn’t grammatically correct, please rephrase this sentence.

Line 117-118: please rephrase the sentence since the diabetes without end organ damage and congestive heart failure also are predominant in women.

Line 197: please correct “simple size” to “sample size”

The authors should consider to include description on how and by whom the death certificates were filled in since this might be performed differently in other countries.

Round 2

Reviewer 3 Report

The authors have satisfactorily responded to all my questions and made the necessary changes to the manuscript. I have no further comments.